# Test and Simulation Analysis of the Working Process of Soybean Seeding Monomer

Dongxu Yan [1], Tianyue Xu [2], Jianqun Yu [3], Yang Wang [3], Wei Guan [4], Ye Tian [5] and Na Zhang [3,*]

1 Hua Lookeng Honors College, Changzhou University, Changzhou 213164, China
2 College of Engineering and Technology, Jilin Agriculture University, Changchun 130022, China
3 School of Biological and Agricultural Engineering, Jilin University, Changchun 130022, China
4 School of Mechanical and Aerospace Engineering, Jilin University, Changchun 130025, China
5 Center of Industry and Technology, Hebei University of Technology Petroleum, Chengde 067000, China
* Correspondence: zna18@mails.jlu.edu.cn; Tel.: +86-13844941571

**Abstract:** Soybean seeding monomers can realize the process of opening, seed throwing, covering, and compacting when they work. Due to the complexity of their working process, the relevant process cannot be analyzed by the discrete element method (DEM) alone. The DEM coupled with the multi-rigid body dynamics method (MBD) can solve the above problem, and the simulation analysis of the above process is realized by coupling the EDEM software with RecurDyn software. The changes in the position of soybean seed particles before and after covering and compacting are analyzed. The results show that when the working speed of the seeding monomer increases, the distance along the vertical direction of the soybean seed particles after covering gradually increases, and the distance along the horizontal direction gradually decreases. The effect of different working speeds of seeding monomer on the opening situation and the variation in seed particle positions is studied. The results show that the ditch angle gradually decreases as the working speed of the seeding monomer increases. The distribution of seed particle spacing is also analyzed. The above tests are simulated, and the results show a high agreement between the simulation and test results, proving the accuracy of the coupling method. This paper applies the coupling method for the first time to the simulation of the seeding monomer. This method can be applied not only to the analysis of the sowing process of soybean seeding monomers, but also be applied to the analysis of other machinery working processes, such as the tillage process, the sieving process, the planting and harvesting processes of crops, etc. It also deepens the application of the discrete element method in the field of agriculture.

**Keywords:** DEM; MBD; coupled simulation; seeding; soybean; seed–soil

## 1. Introduction

Currently, the discrete element method (DEM) has become a common method for analyzing the contact interaction between particles and between particles and mechanical components. It has been widely used in fields such as agricultural engineering [1–9].

Wang [10] analyzed the simulation and test results of the sieving process of a pendulum screen based on the coupled DEM and multi-body dynamics (MBD) algorithms, using AgriDEM software developed independently by the Digital Design Laboratory of the College of Biological and Agricultural Engineering, Jilin University.

Yuan et al. [11] established a self-excited vibration deep loosening machine–soil system model based on the coupling algorithm of DEM and MBD. AgriDEM software was used to simulate and analyze the interaction between mechanical components and soil particles, and the accuracy of the model was verified.

Xu [12] simulated the working process of the coverer and roller based on the coupling algorithm of DEM and MBD, using EDEM software coupled with ADAMS software.

The feasibility and applicability of the coupling method were verified by comparing the displacement of seeds during simulation and testing.

The above analysis shows that the use of different software coupling methods to analyze the field of agricultural machinery has been initially applied. However, for the sowing monomer, when it works under the action of the traction machine, the processes of opening the furrow, throwing the seed, covering, and compacting can be realized. The whole process is a complex mechanical movement. It has not yet been analyzed and studied in depth.

In this paper, test analyses of the covering, compacting, and ditching of seeding monomers are performed. On this basis, an in-depth study of these problems is carried out by means of simulation analysis. The simulation results are compared with the test results to prove the feasibility of the coupling method and the validity of the simulation parameters. It provides a reference for analyzing and optimizing agricultural machinery components such as the opener, cover and roller. The method also offers the possibility for the simulation and analysis of other agricultural machines.

## 2. Composition and Structure of the Seeding Monomer

This paper takes the King Helen soybean seeding monomer as the test object, whose 2D perspective view is shown in Figure 1. The seeding monomer mainly consists of the opener, seed box, seedmeter, coverer, and roller through a combination of connecting devices. The main working parts of the sowing unit are described below.

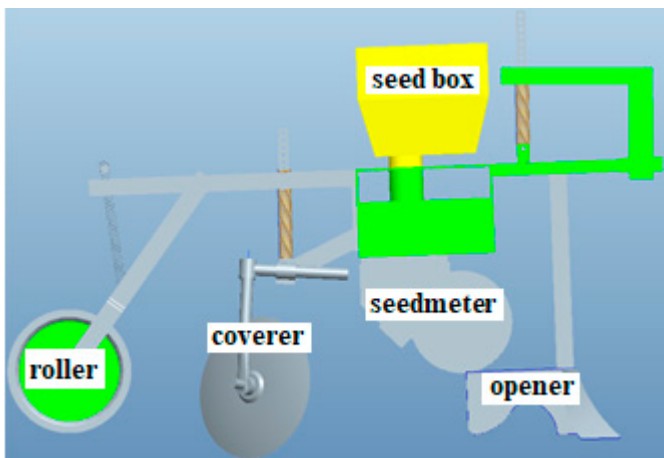

**Figure 1.** Two-dimensional perspective view of the seeding monomer.

### 2.1. Opener

The function of the opener is mainly to open the seed furrow, guide the seeds into the seed furrow and make the wet soil cover the seeds when the seeding monomer is working. Its main structure types are hoe shovel type, wide wing shovel type, arrow shovel type, core share type, slide knife type, double disc type, single disc type, etc. The opener on the seeding monomer of this paper is the core-share type opener, whose structure is shown in Figure 2.

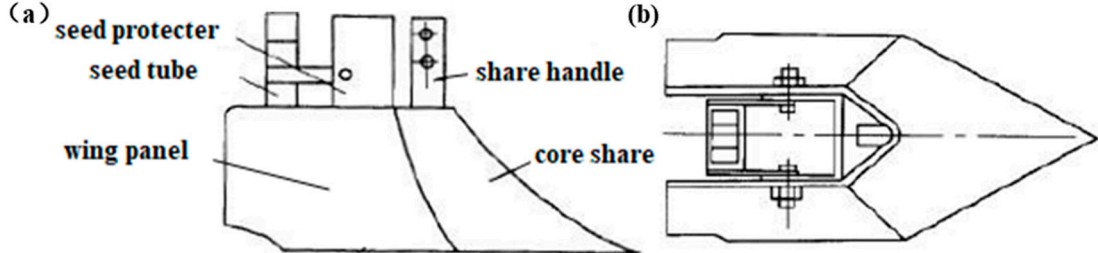

**Figure 2.** (**a**)The front view and (**b**) top view of core-share opener [13].

## 2.2. Vertical Type-Hole Wheel Seedmeter

The vertical type-hole wheel seedmeter mainly consists of seed cylinder, seed rower body, type-hole wheel, seed scraper and seed guard, etc.; see Figure 3.

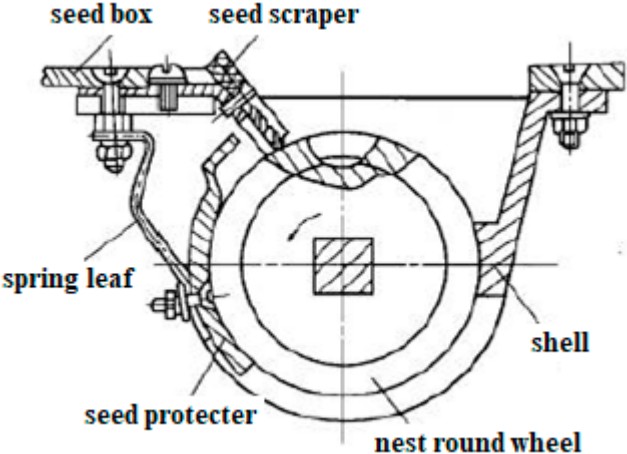

**Figure 3.** Structure diagram of the type-hole wheel seedmeter [13].

### 2.2.1. Type-Hole Wheel

The shape of the profile hole can be cylindrical, conical, and hemispherical. In order to reduce damage to the seed when filling and scraping, generally, the type-hole is paired with a front groove, a tail groove, or a chamfer. Its diameter and depth with the size of the soybean seed are sized to adapt.

In this paper, the seedmeter is a double-row type-hole wheel with a diameter of 200 mm, a hemispherical hole shape, a hole diameter of 8.5 mm, and 100 holes. When the type-hole wheel rotates, the seeds are distributed inside each hole, and the seeds are broadcast with the rotation of the wheel. See Figure 4.

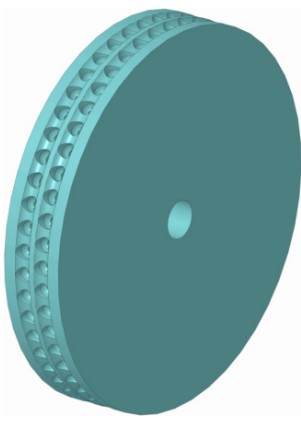

**Figure 4.** The double-row type-hole wheel.

### 2.2.2. Seed Scraper

The seed scraper used on the type-hole wheel seed meter has various forms, and the common ones include brush, rubber seed scraping tongue, brush seed scraping wheel, rubber brush seed wheel, steel knurled seed scraping wheel and so on. The installation position of the seed scraper is indicated by the β angle, generally 22~45°. The seed scraper in this paper is a rubber seed scraper whose installation angle β is 30°. In order to avoid damaging the seeds, the seed scraper cannot be installed vertically above the type-hole wheel, so it should be installed with a certain safety angle α. In this paper, angle α is 10°, see Figure 5.

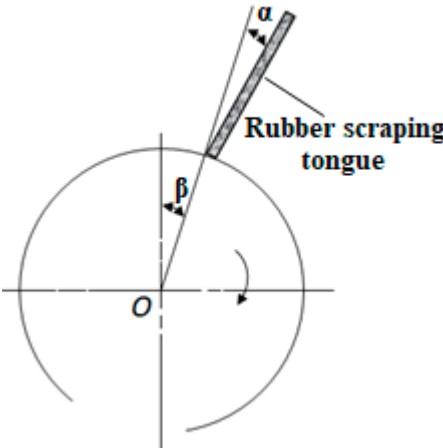

**Figure 5.** The installation angle of rubber scraping tongue [13].

### 2.2.3. Seed Guard Plate

The seed guard is used to ensure that the seed in the type-hole does not fall out from the type-hole during movement so that it reaches the seed-throwing place and is accurately put into the seed furrow. The material of the seed guard is made of iron, plexiglass or foam, and some seed guards are part of the seedmeter shell. In this paper, the seed guard is made of metal tin, which is connected with the seedmeter shell.

### 2.3. Coverer

After the seeds fall into the bottom of the furrow, the furrow opener will cover the seeds with a shallow layer of soil, and then it needs to be covered to make it reach a certain depth of coverage. The type of cover in this paper is a double-disc-type eight-character cover, as shown in Figure 6. The angle of tension of the double-disc cover can be adjusted. When the cover tension angle was varied from 50° to 70°, the congestion of the disc was relieved as the disc tension angle increased, indicating that the larger the tension angle of the cover disc was, the less the soil was disturbed [14]. Based on the above study, the tension angle of the overburden disc is 60° during the simulation.

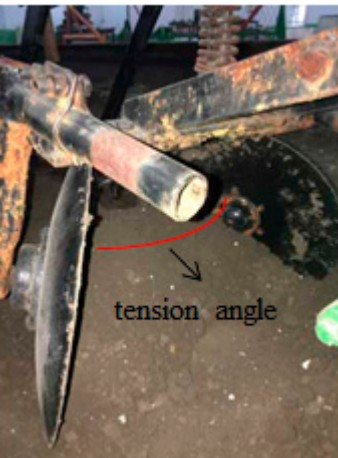

**Figure 6.** Double-disc-type eight-character cover.

*2.4. Compacting Roller*

Compaction is very necessary after covering soil. The main structure types of compacting roller are: a cylindrical compacting roller, concave and convex compacting rollers, a conical compound compacting roller, rubber ring compacting roller, wide compacting roller, narrow rubber compacting roller, etc. The roller of the seeding monomer in this paper is a cylindrical compacting roller with a diameter of 260 mm and a working width of 1000 mm, as shown in Figure 7.

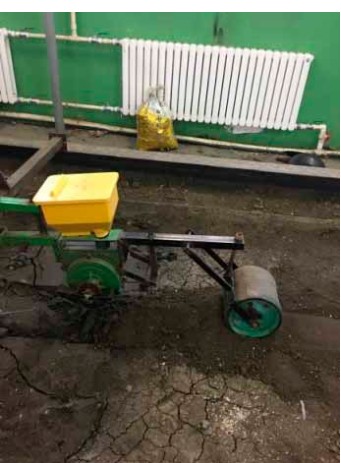

**Figure 7.** Cylindrical compacting roller.

## 3. Covering and Compacting Test

The tests are conducted in the soil tank test bed of the agricultural laboratory of Jilin University. The soil type is a sandy loam with a moisture content of $18 \pm 2\%$. The soil tank test vehicle is connected to the seeding monomer; see Figure 8a. To construct the seed furrow artificially, two sets of seed furrows with a height of 60 mm, a width of 150 mm and a length of 4000 mm are constructed on the surface of the seed bed, and the seed trench appears in the middle, as shown in Figure 8b. The seeds are sown by hand with a distance of 300 mm between every two seeds. The place of the seeds is marked on a string on the outside of the seed trench; see Figure 8c,d. The vertical distance of the seeds is also measured, as shown in Figure 8e.

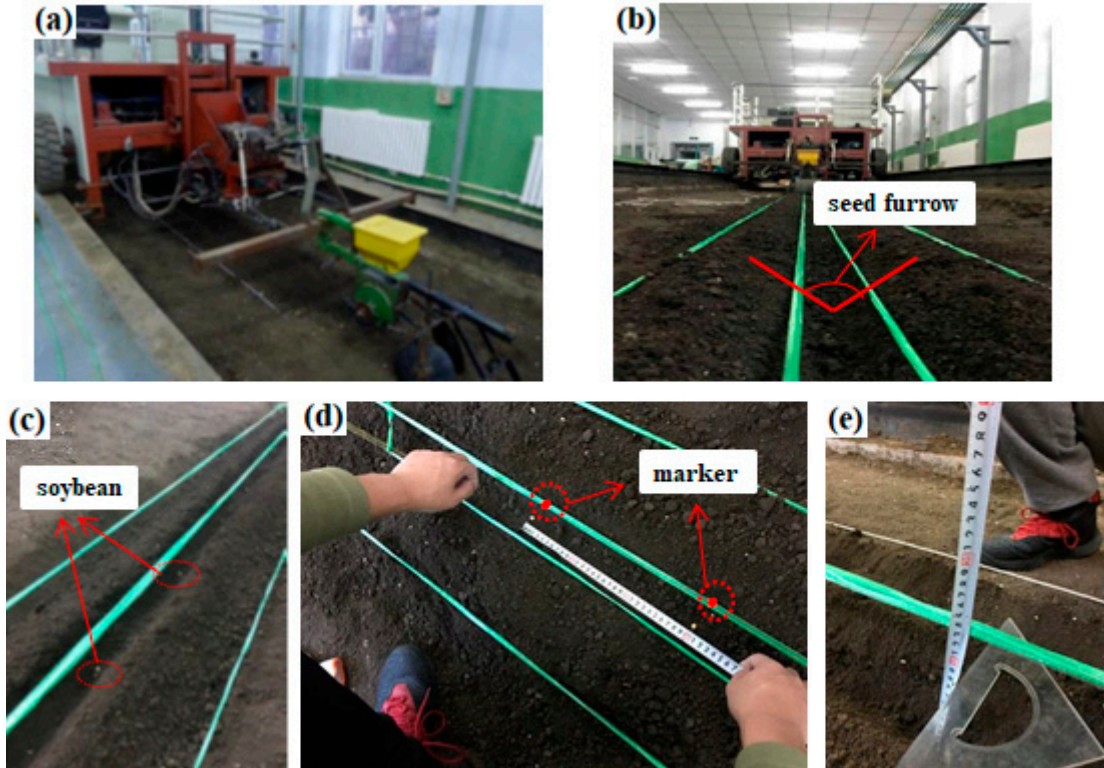

**Figure 8.** The pictures of the covering and compacting test process, (**a**) the seeding monomer, (**b**) the seed furrow, (**c**) soybean seed particles seeded in the seed furrow, (**d**) the measurement of seed particle spacing and (**e**) the measurement of vertical distance of soybean seed particles.

The covering test is carried out as follows: firstly, only the cover of the sowing unit is operated, the covering disc is tensioned at an angle of 60°, and the test vehicle is driven at a speed of 0.75 m/s, 1.11 m/s and 1.47 m/s. After covering, the soil covering the seeds is removed, and the horizontal and vertical distances of the seeds are measured. The change in the position of the soybean seed particles after mulching is analyzed, and the test data are recorded.

The compacting test was carried out as follows, with the cover and compactor of the sowing monomer working simultaneously and the cover disc tensioned at an angle of 60°. After covering and compacting the seeds, the soil was removed from the top of the seeds, and the horizontal and vertical distances of the seeds were measured. The change to the position of the soybean seed particles after compacting was analyzed, and the test data were recorded.

## 4. Covering and Compacting Simulation

To simulate the working process of the seeding monomer during the covering and compacting test, the coupling between EDEM and RecurDyn needs to be used. The simulation setup steps are as follows: import the STL file of the 3D diagram into RecurDyn software; set the material property of the part to "Steel"; and then set the connection method between the rigid bodies involved in the seeding monomer (see Table 1).

**Table 1.** Connection between rigid bodies in the seeding monomer.

| Constraint Number | Constraint Object | Constraint Type |
|:---:|:---:|:---:|
| 1 | opener–ground | sliding pair |
| 2 | seeding wheel–shaft | revolute pair |
| 3 | coverer–shaft | revolute pair |
| 4 | cross beam–back beam | revolute pair |
| 5 | back beam–roller | revolute pair |

In order to realize the coupling of the two software types, it is necessary to set up the corresponding settings in EDEM (Version 2018). After opening the EDEM software, it is necessary to select RecurDyn Coupling in the coupling option of EDEM. Next, the wall files are imported into EDEM. The height of the soil tank is set according to the height of the seeding monomer. After analysis, the dimensions of the soil tank are determined as 4820 mm × 820 mm × 420 mm. The soybean seed variety in this paper is SN42 and has an ellipsoidal shape, the DEM model of soybean seed particles was established using the 13-sphere model, and a population of soybean seed particles was generated in the simulation according to a normal distribution [15,16]. The soil particle shapes are sphere-like and triangle-like. In order to save simulation time, the soil particle model was simulated with a particle size of 10 mm and a population of soil particles was generated according to a uniform distribution. The physical and mechanical parameters of soybean seed particles, soil particles and the material of the seeding monomer (galvanized steel) are determined by the author's period research, as shown in Table 2 [15,16].

**Table 2.** Physical and mechanical parameters of the simulated materials [15,16].

| Material | Density, kg/m$^3$ | Poisson's Ratio | Elasticity Modulus, Pa |
|:---:|:---:|:---:|:---:|
| Soybean | 1257 | 0.4 | $7.60 \times 10^8$ |
| Galvanized steel | 7850 | 0.3 | $7.90 \times 10^{11}$ |
| Soil | 1844 | 0.25 | $1.00 \times 10^6$ |

In order to save calculation time, the radius of soil particles is taken as 10 mm for the simulation [17], and about 200,000 soil particles are generated according to the uniform size. The contact model is the Edinburgh model [18–21].

At the time of simulation, the soybean variety is SN42. The parameters of the material interactions during the simulation are measured. In this paper, the particle size becomes larger during simulation, and the surface energy between particles is adjusted to 4 by simulation analysis. The specific parameters are shown in Table 3 [15,16].

**Table 3.** Parameters of material interactions [15,16].

| Parameter | SN42–SN42 | SN42–Soil | Soil–Soil |
|:---:|:---:|:---:|:---:|
| Coefficient of restitution | 0.627 | 0.75 | 0.6 |
| Coefficient of static friction | 0.2 | 0.254 | 0.9 |
| Coefficient of rolling friction | 0.02 | 0.011 | 0.7 |
| Constant pull-off force, N | 0 | 0 | 0 |
| Surface energy, J/m$^2$ | 0 | 0.5 | 4 |
| Contact Plasticity Ratio | 0.35 | 0.35 | 0.35 |
| Slope Exp | 1 | 1.5 | 1.5 |
| Tensile Exp | 1 | 1 | 1 |
| Tangential stiff Multiplier | 0.67 | 0.67 | 0.67 |

The coefficient of restitution, the coefficient of static friction, and the coefficient of rolling friction between the soil particles and the soil tank and the soybean seed particles and the soil tank have little effect on the simulation results in this paper. Therefore, the values are taken by the system as default.

Figure 9 shows a screenshot of the simulation interface of the two software types at different simulation moments for a seeding monomer working speed of 1.47 m/s.

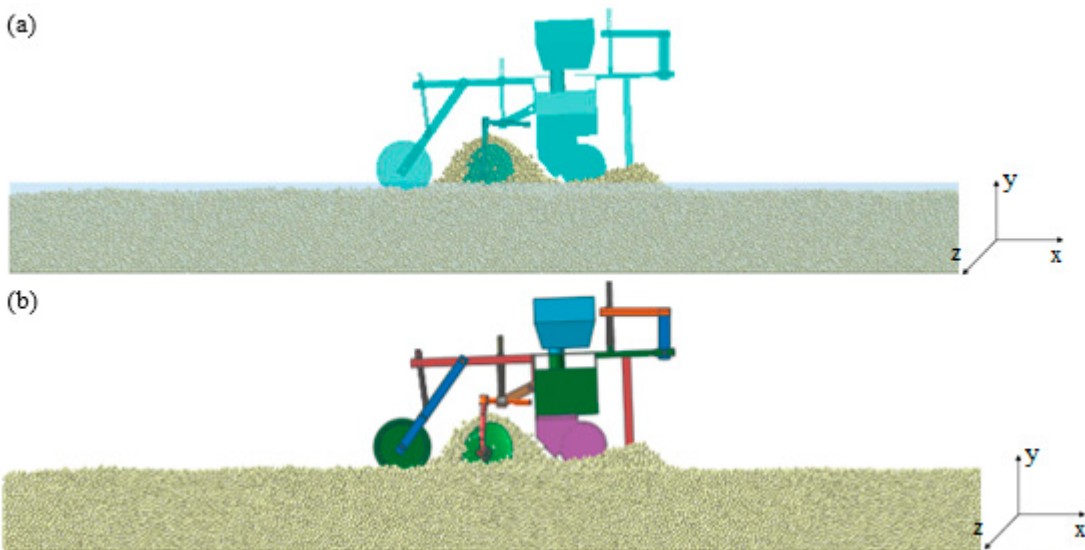

**Figure 9.** Screenshot of (**a**) EDEM simulation and (**b**) RecurDyn simulation at t = 2 s.

## 5. Analysis of the Results

Analysis and statistics of the changes in the position of soybean seed particles after covering and compacting, the opening of the furrow opener and the distribution of grain spacing at different working speeds were carried out.

### 5.1. Analysis of Covering and Compacting Results

The change in the position of soybean seed particles after covering is compared with the test results. When conducting the simulation, the +X direction was set as the horizontal driving direction of the seeding monomer, the Z-axis direction corresponded to the lateral displacement of the test, and the *Y*-axis direction corresponded to the vertical direction.

The analysis process for the variation in horizontal and vertical displacements of a seed particle is taken as an example. The position of the soil particle without covering is shown in Figure 10a. Figure 10b shows the position of the soybean seed particle when it has been covered with soil. Figure 10c shows the position of the soybean seed particles in the soil after compacting. Zooming in on the images, it can be clearly observed that the soybean seed particles are significantly displaced in the *X*-axis direction, as shown in Figure 10d. Five soybean seed particles are selected to obtain information on the position of soybean seed particles in the X, Y and Z directions, and the test data are analyzed.

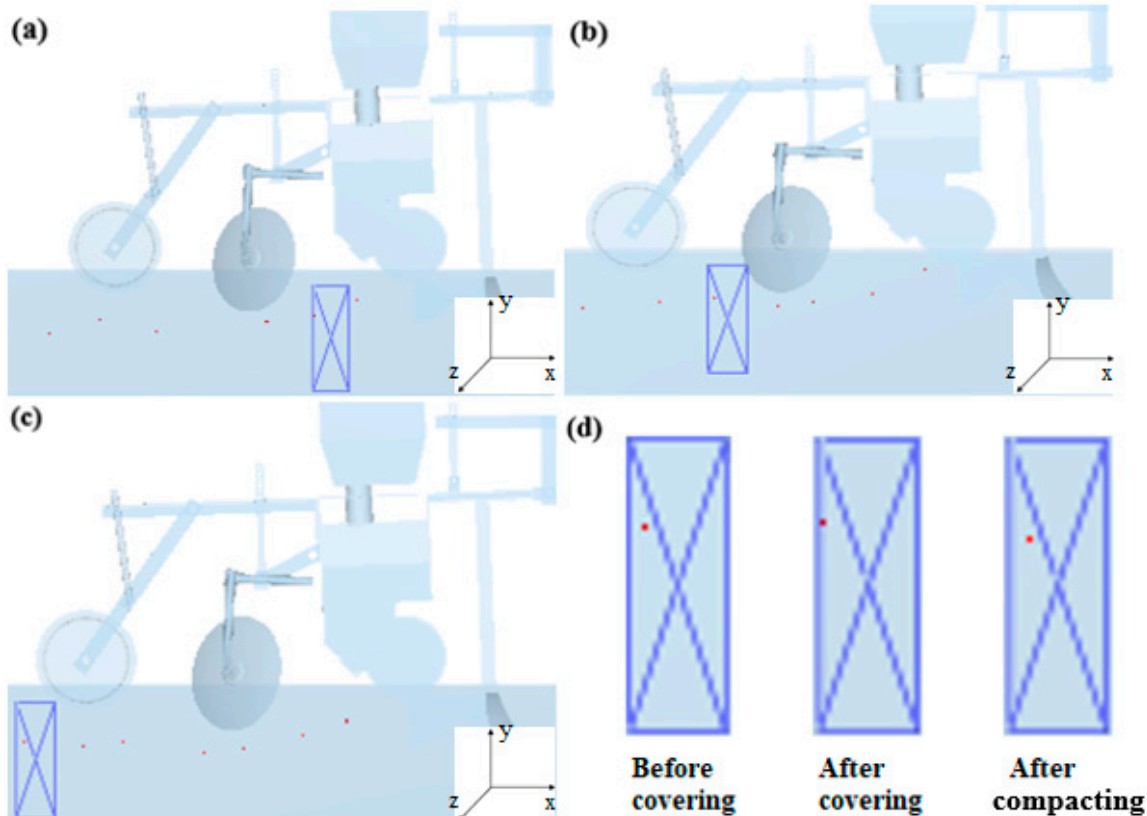

**Figure 10.** Position of soybean seed particles in the soil (**a**) before covering, (**b**) after covering, (**c**) after compacting and (**d**) partially enlarged view.

5.1.1. Analysis of Covering Test Results

The simulation and test results [14] of the change in the vertical and horizontal position of soybean seed particles before and after covering at different working speeds of the seeding monomer are shown in Figure 11. The bars in the figure are the error band resulting from the processing of the replicate test and simulation results and are expressed as the standard deviation. The analysis shows that when the working speed of the seeding monomer increases, the distance along the vertical direction of the soybean seed particles after covering gradually increases, and the distance along the horizontal direction gradually decreases. The simulation results have the same trend as the test results as a whole.

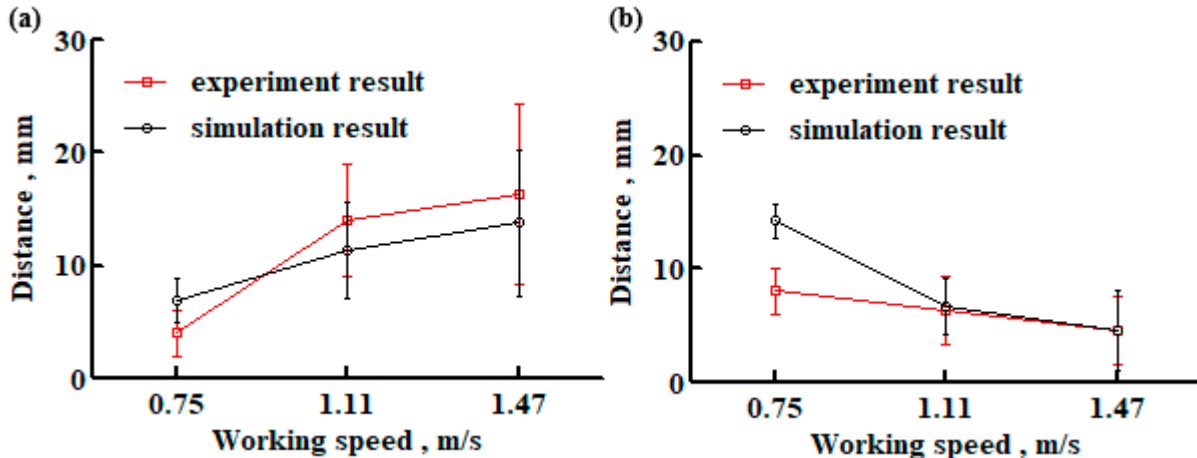

**Figure 11.** The relationship between the change in the position of soybean seed particles after covering and the working speed of the seeding monomer of (**a**) vertical and (**b**) horizontal.

### 5.1.2. Analysis of the Results of the Compacting Test

The simulation and test results [14] of the change in the vertical and horizontal position of soybean seed particles before and after compacting at different working speeds of the seeding monomer are shown in Figure 12. Analysis shows that with the increase in the working speed of the seeding monomer, the change in the position of soybean seed particles in both horizontal and vertical directions after compacting shows a trend of gradually becoming larger. When the working speed of the seeding monomer is 1.11 m/s, the horizontal and vertical position changes in soybean seed particles differ the most from the test results, and the difference between them and the test is 1.69 mm and 1.33 mm, respectively, which is a relatively small value. Therefore, it can be considered that the overall simulation results and the test results are relatively close to each other.

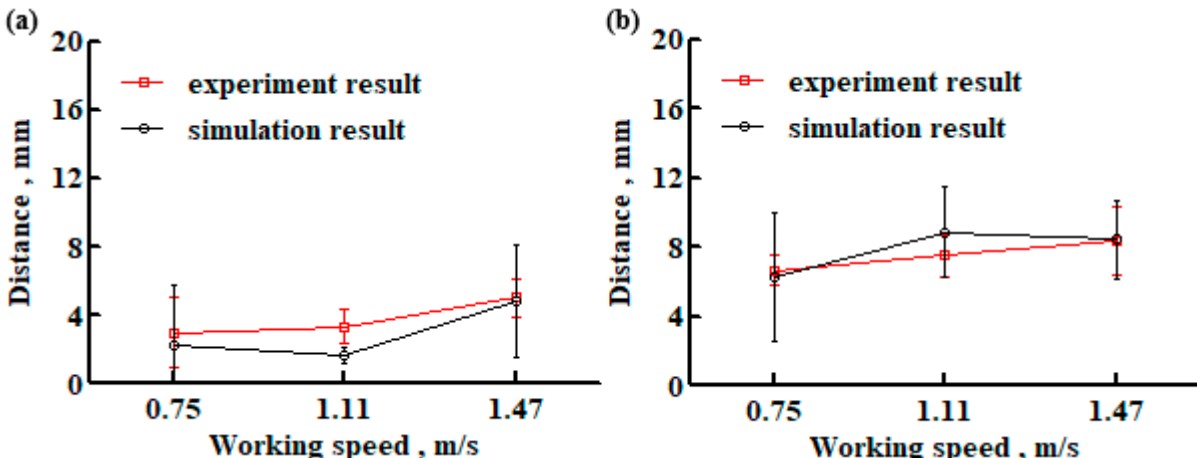

**Figure 12.** The relationship between the change in the position of soybean seed particles after compacting and the working speed of the seeding monomer of (**a**) vertical and (**b**) horizontal.

The change in the position of soybean seed particles in the vertical direction after covering and after compacting at different working speeds is shown in Figure 13. It can be seen from the analysis that the change in the vertical direction of soybean seed particles in the soil after covering and after compacting is not significant. The maximum difference between the two is 3.7 mm when the working speed of the seeding monomer is 0.75 m/s. The minimum difference between the two is 1.77 mm when the working speed of the seeding monomer is 1.11 m/s.

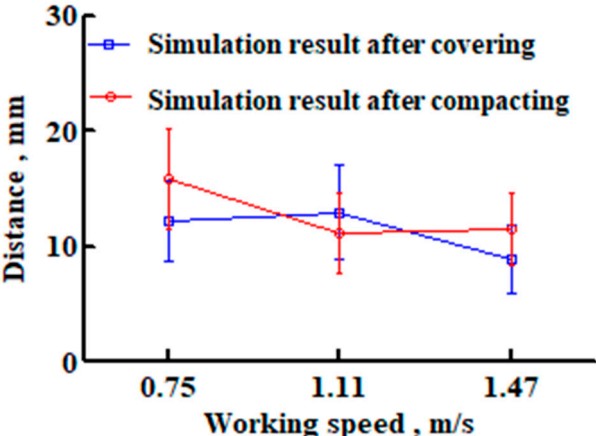

**Figure 13.** The change in the position of soybean seed particles in the vertical direction after covering and after compacting at different working speeds.

The comprehensive analysis shows that the faster the working speed of the seeding monomer, the greater the change in the position of the soybean seed particles that occurs. The reason for this is that the faster the seeding monomer works in contact with the soil particles, the more kinetic energy the soil particles will have, which means that the soil particles will be displaced more when they come to rest. The seeding monomer coupling simulation results have the same trend as the test results, and the parameters chosen for the coupling simulation are reasonable.

### 5.2. Analysis of the Open Furrow Results

The ditch angle θ of the opener when the working speed of the seeding monomer is 1.47 m/s, as shown in Figure 14. The red particle in the figure is the SN42 soybean seed particle that is falling down.

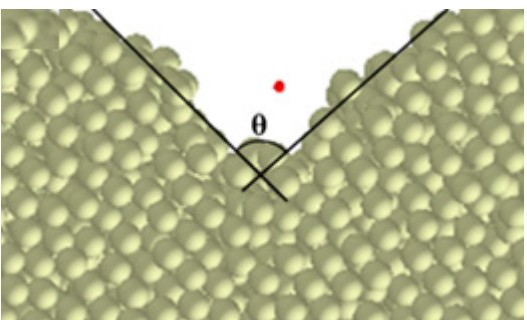

**Figure 14.** The ditch shape diagram at a seeding monomer working speed of 1.47 m/s.

The ditch angle is analyzed at the simulation moment of 1 s, 2 s and 3 s when the working speed of the seeding monomer is 0.75 m/s, and the average value is obtained. The ditch angle of the soil at working speeds of 0.75 m/s, 1.11 m/s and 1.47 m/s are analyzed separately, and the results are shown in Figure 15. The analysis shows that the ditch angle tends to decrease gradually as the working speed of the seeding monomer increases within the range studied in this paper. When the working speed is 0.75 m/s, the maximum ditch angle is 94.5°. When the working speed of the seeding monomer is 1.47 m/s, the ditch angle is the smallest, and the value is 90.31°.

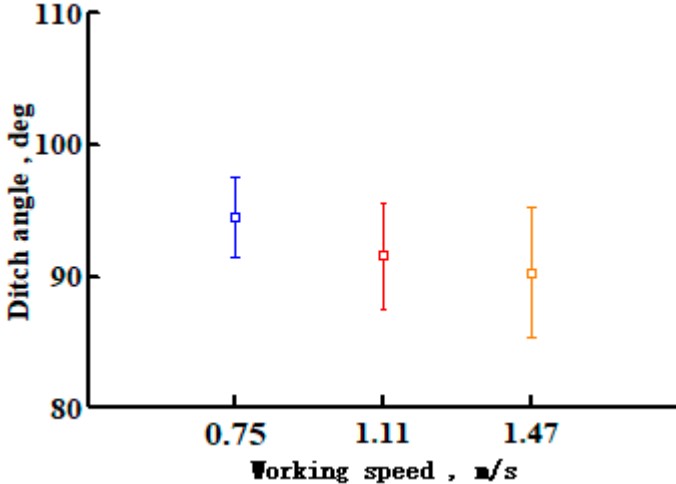

**Figure 15.** The ditch angle of the soil at different working speeds.

From the above analysis, it can be seen that the ditch angle gradually decreases as the working speed of the seeding monomer increases. The likely reason for this is that the faster the seeding unit works in contact with the soil particles, the greater the kinetic energy

of the soil particles, which results in more soil particles being distributed on the outside of the seed furrow.

### 5.3. Seed Spacing Analysis

In the simulation, a 10 mm × 10 mm × 10 mm particle factory is set up at the seed discharge port of the seedmeter, and the soybean seed particles are generated in the particle factory at a rate of 10 per s. The speed of the particle factory is the same as that of the seeding monomer. Therefore, the theoretical seed spacing for soybean seeding, corresponding to the forward speed of the seeding monomer, was 75 mm, 110 mm, and 147 mm for 0.75 m/s, 1.11 m/s, and 1.47 m/s, respectively.

The distribution of soybean seed particles in the soil after covering and compacting the seeds at different working speeds is shown in Figures 16–18.

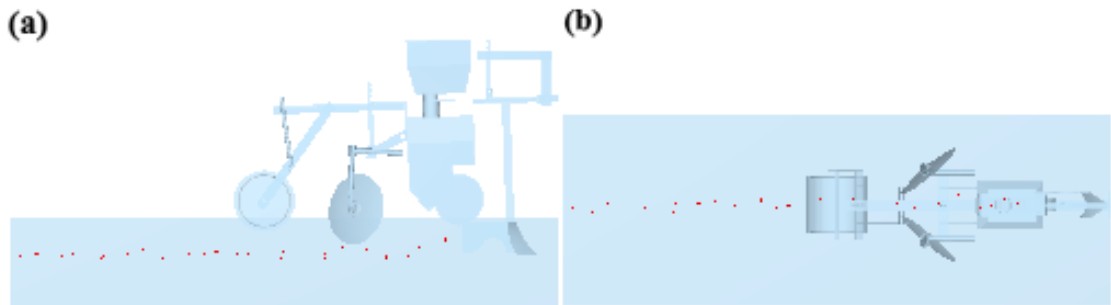

**Figure 16.** Distribution of soybean seed particles in the soil in (**a**) XOY plane and (**b**) XOZ plane at a working speed of 0.75 m/s.

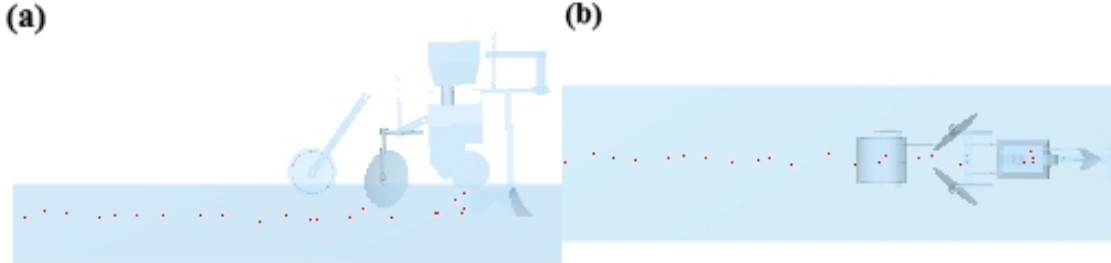

**Figure 17.** Distribution of soybean seed particles in the soil in (**a**) XOY plane and (**b**) XOZ plane at a working speed of 1.11 m/s.

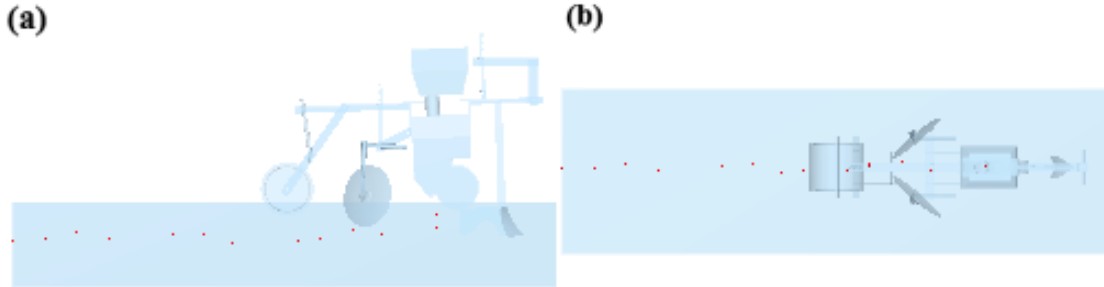

**Figure 18.** Distribution of soybean seed particles in the soil in (**a**) XOY plane and (**b**) XOZ plane at a working speed of 1.47 m/s.

Figure 19 shows the difference between the theoretical seed spacing and the simulation results at different working speeds. For the sake of comparison, the concept of relative error is defined. The relative error is the ratio of the difference between the mean value of the simulation results and the theoretical value. When the working speed is 0.75 m/s, the

difference between the simulation result and the theoretical seed spacing is only 0.48 mm, with a relative error of 0.6%. When the working speed is 1.11 m/s, the difference between the simulation result and the theoretical seed spacing is the largest, and the difference is 6.44 mm, with a relative error of 5.8%. When the working speed is 1.47 m/s, the difference between the simulation result and the theoretical seed spacing is 3.42 mm, and the relative error is 2.3%. The comprehensive analysis shows that the simulation results are basically consistent with the theoretical grain distance at different working speeds.

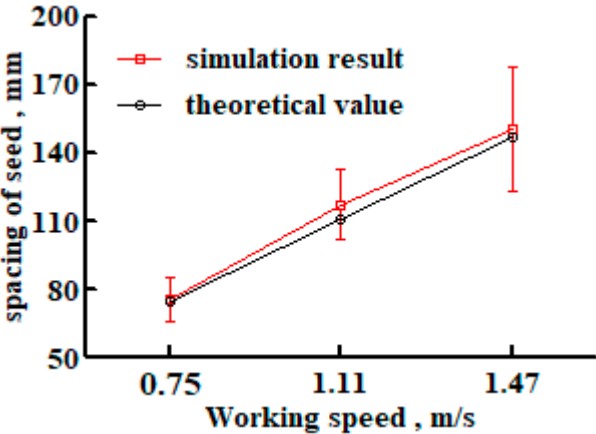

**Figure 19.** The relationship between seed spacing and working speed.

## 6. Conclusions

In this paper, tests on the covering and compacting processes of seeding monomers are carried out, and the above processes are simulated and analyzed by using the DEM coupled with MBD. Simulation analysis is also carried out for the open furrow situation and the uniformity of seed spacing. In the future, DEM and MBD coupling will be widely used for the optimal design of earth-touching machine components. The specific conclusions are as follows:

(1) The EDEM and RecurDyn software are coupled to simulate the process of opening, seed throwing, covering, and compacting of the seeding monomer. The comparison of simulation and test results shows that with the increase in working speed, after covering, the position change in soybean seed particles in the vertical direction gradually increases and in the horizontal direction gradually decreases.

(2) After compacting, the position change of soybean seed particles in the vertical and horizontal directions gradually increases. The simulation is basically consistent with the trend of the test results. Meanwhile, after the experience of covering the soil and then compacting it, the change in the vertical direction of soybean seed particles is not significant.

(3) The ditching angle of the soil gradually decreases when the working speed of the seeding monomer increases. The average seed spacing of the simulation is basically the same as the theoretical spacing at different working speeds of the seeding monomer.

(4) In the simulations of this paper, the boundary conditions of the mechanical components have been ignored, which may produce certain errors and, therefore, will need to be taken into account in future research work. The influence of different soil particle models on the simulation results also needs further study.

**Author Contributions:** Conceptualization, D.Y.; methodology, D.Y.; validation, D.Y. and W.G.; investigation, W.G. and resources, J.Y. and W.G.; writing—original draft preparation D.Y. and N.Z.; writing—review and editing, Y.W. and N.Z.; supervision, Y.T. and T.X.; project administration, J.Y.; funding acquisition, J.Y. All authors have read and agreed to the published version of the manuscript.

**Funding:** The authors are grateful to the National Natural Science Foundation of China (No. 52130001) for the financial support of this work.

**Data Availability Statement:** Not applicable.

**Conflicts of Interest:** The authors declare no conflict of interest.

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
