# Peer review of "Test and Simulation Analysis of the Working Process of Soybean Seeding Monomer"

_agriculture, doi:10.3390/agriculture12091464_

Round 1

Reviewer 1 Report (Previous Reviewer 1)

The modifications made in this paper are very appropriate.

The scientific quality has improved a lot.

Thanks for accepting my suggestions.

Author Response

thankyou!

Reviewer 2 Report (New Reviewer)

The manuscript presented for review is an extension of the work done in [14]. They include simulation of the processes of opening, seed throwing, covering and compacting by coupling the EDEM and RecurDyn software. The authors studied the effect of different working speeds of seeding monomer on the opening situation, the variation of seed particle position and the changes in the position of soybean seed particles before and after covering and compacting as well as the uniformity of seed spacing between seeds. They also compared the simulation results with experimental results to prove the feasibility of the coupling method and the validity of the simulation parameters. The results of the study can provide some reference for the development of agricultural machinery components.

The research carried out by the Authors, the methods used and their description do not raise any major objections. However, some issues require supplementation or correction, namely:

1- None of the figures that present data explain what the bars are.

2- Something is missed in sentences in lines 78 and 81.

3- In fig 5, the authors explain the installation angle \beta but not \alpha.

4- In line 273 the authors mention “the theoretical seed spacing”. What theory are you referring to?

5- Line 84: “diameter of 8.5 mm, and 100 holes.” What for? It seems to me that they have to clarify if the holes are to host the seeds or something else.

6- Line 126: What do you mean by “soil monopolies”?

Author Response

Response to Reviewer 2 Comments

Point 1: None of the figures that present data explain what the bars are.

Response 1:  The bars is the error band resulting from the processing of the replicate test and simulation results and is expressed as the standard deviation.According to the reviewer’s comment, the author have described it, as detailed in line 222-224.

Point 2: Something is missed in sentences in lines 78 and 81.

Response 2: According to the reviewer's comments, the author has revised the incoherent phrases, as detailed in line 89-91.

Point 3: In fig 5, the authors explain the installation angle \beta but not \alpha.

Response 3: According to the reviewer's comments, the authors explain the angle α. In order to avoid damaging the seeds, the seed scraper cannot be installed vertically above the type-hole wheel, so it should be installed with a certain safety angle α, in this paper angle α is 10°,as detailed in line 102-105.

Point 4: In line 273 the authors mention “the theoretical seed spacing”. What theory are you referring to?

Response 4: The theoretical seed spacing is the calculated result. Authough the simulations are usually formulated based on the theoretical aspects. However, several physical simulations are carried out under the same conditions. The results of each simulation are also different. According to the reviewer’s comments, the author elaborate on the theoretical values, as detailed in line 288-290. 

Point 5: Line 84: “diameter of 8.5 mm, and 100 holes.” What for? It seems to me that they have to clarify if the holes are to host the seeds or something else.

Response 5: When the type-hole wheel rotates, the seeds are distributed inside each hole, and the seeds are broadcast with the rotation of the wheel. According to the reviewer's comments, the role of the holes is explained in the paper, as detailed in line 94-96.

Point 6: Line 126: What do you mean by “soil monopolies”?

Response 6: According to the reviewer's comments, the meaning of “soil monopolies” is ambiguous.

Therefore, in this paper, we replaced the soil monopolies with a seed furrow, as detailed in line 138.

Reviewer 3 Report (New Reviewer)

Although the work is of interest and a new application of DEM coupled with MBD, first and foremost, there are major comments to consider listed below.

Abstract

-This section should be restructured in view of problem statement, objectives, detailed information related to the adapted methodology in this research study, the best findings/key data of this study as well as novelty/originality of this study should be clarified carefully.

-Future aspects/recommendations should be included. The proposed applications are also essential to be mentioned.

Introduction

- Literature review is subpar.

- Novelty/originality is drawn poorly and does not meet the requirements of reputed journals.

- The authors should clarify the research gap after the literature review since the significance and novelty of this work are vague. So, Introduction and literature review should be improved. And accordingly, the references should be added.

- References No. 1 and 3 are not in the field of agricultural engineering. Instead of those, the following references are more appropriate to the application:

·         (2006). The discrete element method (DEM) to simulate fruit impact damage during transport and handling: Model building and validation of DEM to predict bruise damage of apples. Postharvest Biology and Technology.

·         (2019). Two-dimensional particle shapes modelling for DEM simulations in engineering: A review. Granular Matter.

·         (2021). Modelling and simulation of fruit drop tests by discrete element method. Biosystems Engineering.

 - In the literature review, more internationally references should be cited.

- And some relevant-papers to particle shape modelling could be added such as:

·         Two-dimensional particle shapes modelling for DEM simulations in engineering: A review. (2019). Granular Matter

·         An Approach to represent realistic particles of bulk assembly in three-dimensional-DEM simulations and applications. Commun. Agric. Appl. Biol. Sci. 76(1), 33–36 (2011)

·         A novel approach to a realistic discrete element modelling (DEM) in 3D. Commun. Agric. Appl. Biol. Sci. 72(1), 205–208 (2007)

·         Shape modelling of fruit by image processing. Commun. Agric. Appl. Biol. Sci. 70(2), 161–164 (2005)-

- There is a lack of critical discussion in this section.

- Nomenclature is necessary for symbols and abbreviations in an appropriate section based on journal guide.

Material and methods

-This section should be rewritten clear enough that other interested researchers could do the same.

- Figure and table captions should be clear and precise that are understandable without the text.

- Figures must be presented in good quality and accurate.

- What about particle shape of soil and seed? What generation method was used?

- What about distribution of soil and seed size and shape?

- What about theoretical/simulation formulation?

- How coupling EDEM and RecurDyn works? What version of the software was used? There is no detail in the text.

- In Table 2 and 3, how the procedure for the measurement of those values? All details should be included.

- The considered boundary condition should be explained in details in view of the impact of accuracy and real practice.

- Validation, evaluation and calibration should be clarified, since the results should be supported with valid data. And the Experimental, Validation and Measurement systems are unclear. They should be clearly in details that any researchers could do the same.

Results and discussion

- The results should be discussed in comparison to other similar research topics/findings, if possible.

The scientific reasons for the findings should be explained in more detail.

- The simulations are usually formulated based on the theoretical aspects. So, what do you mean by the following sentence: “Figure 19 shows the difference between the theoretical seed spacing and the simulation results at different working speeds”.

- What is the relative error of 0.6%? It should clearly be explained.

- What about vertical and horizontal bias of seed planting?

- What about distribution of seeds size and shape?

Conclusions

The future works, pros and cons of the current work should clearly be described.

Author Response

Response to Reviewer 3 Comments

Point 1: Abstract

-This section should be restructured in view of problem statement, objectives, detailed information related to the adapted methodology in this research study, the best findings/key data of this study as well as novelty/originality of this study should be clarified carefully.

-Future aspects/recommendations should be included. The proposed applications are also essential to be mentioned.

Response 1: According to the reviewer’s comments, The author has adapted and changed the abstract section of the paper. The novelty and originality of the work is highlighted and the future application of the paper is proposed, as detailed in line 21-35.

Introduction

Point 2: - Literature review is subpar.

- Novelty/originality is drawn poorly and does not meet the requirements of reputed journals.

- The authors should clarify the research gap after the literature review since the significance and novelty of this work are vague. So, Introduction and literature review should be improved. And accordingly, the references should be added.

- References No. 1 and 3 are not in the field of agricultural engineering. Instead of those, the following references are more appropriate to the application:

(2006). The discrete element method (DEM) to simulate fruit impact damage during transport and handling: Model building and validation of DEM to predict bruise damage of apples. Postharvest Biology and Technology.

 (2019). Two-dimensional particle shapes modelling for DEM simulations in engineering: A review. Granular Matter.

 (2021). Modelling and simulation of fruit drop tests by discrete element method. Biosystems Engineering.

In the literature review, more internationally references should be cited.

- And some relevant-papers to particle shape modelling could be added such as:

Two-dimensional particle shapes modelling for DEM simulations in engineering: A review. (2019). Granular Matter

An Approach to represent realistic particles of bulk assembly in three-dimensional-DEM simulations and applications. Commun. Agric. Appl. Biol. Sci. 76(1), 33–36 (2011)

A novel approach to a realistic discrete element modelling (DEM) in 3D. Commun. Agric. Appl. Biol. Sci. 72(1), 205–208 (2007)

Shape modelling of fruit by image processing. Commun. Agric. Appl. Biol. Sci. 70(2), 161–164 (2005)-

Response 2: The references cited in the literature review section have been organised according to the comments of the reviewer, and the appropriate references have been cited, as detailed in the references section.

Point 3: There is a lack of critical discussion in this section.

Nomenclature is necessary for symbols and abbreviations in an appropriate section based on journal guide.

Response 3: According to the reviewer’s comments, critical discussions have taken place and the abbreviation of terms has been corrected,as detailed in line 54-66.

Material and methods

Point 4: This section should be rewritten clear enough that other interested researchers could do the same.

Figure and table captions should be clear and precise that are understandable without the text.

Figures must be presented in good quality and accurate.

Response 4: The content of the paper as well as the figure and table captions were revised in accordance with the comments of the reviewer.

Point 5 : What about particle shape of soil and seed? What generation method was used?

- What about distribution of soil and seed size and shape?

- What about theoretical/simulation formulation?

Response 5: According to the authors' previous research, the soybean seed pellet variety in the article is SN42 (sphericity of 94.78%) and has an ellipsoidal shape. A DEM model of soybean seed particles was established using the 13-sphere model, and a population of soybean seed particles was generated in the simulation according to a normal distribution. The soil particle shapes are sphere-like and triangle-like. In order to save simulation time, the soil particle model was simulated with a particle size of 10 mm and a population of soil particles was generated according to a uniform distribution[15], as detailed in line 174-180.  

Point 6: How coupling EDEM and RecurDyn works? What version of the software was used? There is no detail in the text.

Response 6: In order to realize the coupling of the two software, it is necessary to set up the cor-responding settings in EDEM. After opening the EDEM software, it is necessary to select RecurDyn Coupling in the coupling option of EDEM. Next, the wall files are imported into EDEM.

The EDEM's version is 2018, as detailed in line 170-172.

Point 7:  In Table 2 and 3, how the procedure for the measurement of those values? All details should be included.

Response 7: Thank you for your comments. Due to the author's mistake, the sources of the parameters in Tables 2 and 3 have not been described. In fact all the above parameters were measured by the authors in a previous study. Therefore they have not been further elaborated in this paper and in view of this, the corresponding references to the sources of Tables 2 and 3 have been cited in this paper, as detailed in line 183-191.

Point 8: The considered boundary condition should be explained in details in view of the impact of accuracy and real practice.

Response 8: During the tests in the actual soil tank, the interaction between the soil particles and the tank boundary had less influence on the test results and therefore the simulation ignored the interaction between the tank and the soil particle model. Similarly, this paper focuses on the interaction between the particles and therefore ignores the interaction between the soil particles and the mechanical components. Hence, I am really sorry that the paper does not delve into the above issues.

Point 9: Validation, evaluation and calibration should be clarified, since the results should be supported with valid data. And the Experimental, Validation and Measurement systems are unclear. They should be clearly in details that any researchers could do the same.

Response 9: According to the reviewer’s comments, the authors have re-proofed the paper and have made the appropriate changes in the text.

Results and discussion

Point 10: The results should be discussed in comparison to other similar research topics/findings, if possible.The scientific reasons for the findings should be explained in more detail.

Response 10: The reviewer's comments were quite appropriate. This paper focuses on the test and simulation analysis of the working process of a seeding monomer. In fact, it is an original work to simulate the working process of sowing monomer using coupling method. There is a paucity of relevant references for the aforementioned study and therefore the authors have not found corresponding tests for comparative analysis and discussion. The analysis of the results and discussion section of the article has been scientifically analyzed and interpreted in accordance with the comments of the reviewer experts,as detailed in line 262-264 and 282-285.

Point 11: The simulations are usually formulated based on the theoretical aspects. So, what do you mean by the following sentence: “Figure 19 shows the difference between the theoretical seed spacing and the simulation results at different working speeds”.

Response 11: The theoretical seed spacing is the calculated result. Authough the simulations are usually formulated based on the theoretical aspects. However, several physical simulations are carried out under the same conditions. The results of each simulation are also different. According to the reviewer’s comments, the author elaborate on the theoretical values, as detailed in line 292-294.  

Point 12:What is the relative error of 0.6%? It should clearly be explained.

Response 12: The relative error is the ratio of the difference between the mean value of the simulation results and the theoretical value to the theoretical value. According to the reviewers' comments, the authors have elaborated on the relative error in the paper, as detailed in line 307-309.

Point 13:What about vertical and horizontal bias of seed planting?

Response 13: As shown in figure (a), the vertical deviation is the distance at which the seed particles vary in the depth direction (y direction). The horizontal deviation is the distance at which the seed particles vary in the xz plane.

Point 14: What about distribution of seeds size and shape?

Response 14: The soybean variety selected for this paper was SN soybean seed particles. According to a previous study by the authors, its shape was ellipsoid (sphericity of 94.78%) and the population of particles was generated according to a volume normal distribution. Where the standard deviation of the volume normal distribution was obtained after triaxial size analysis of 200 soybean seed particles. Based on the comments of the reviewer,the authors have made a presentation in the paper, as detailed in line 174-180.

Conclusions

Point 15:  The future works, pros and cons of the current work should clearly be described.

Response 15: According to the reviewer’s comments, the authors summarised the gaps in the paper and clearly described the future works, as detailed in line 340-343.

Round 2

Reviewer 3 Report (New Reviewer)

Although the manuscript was improved, there are still some previously mentioned items in the revised version. Those are many minor editorial and technical faults such as:

-    Line 174 needs a reference instead of “authors' previous research“.

-    Reference(s) should be included to Table caption(s) in Table 2 and 3.

-    Some figures are not in good quality, for instant Figure 3.

-    References should be rewritten correctly and in the same format. For example, No.1 (Van Zeebroeck, M.; ...); No.2 (Kafashan, J.; ...); No. 3 (incomplete the author name and format matter); and No. 7, 8, 9 (Yan, D;). No. 15 again. And using “, et al.” for No. 12, 14, 18.

-    Yet, other relevant references could be cited and added.

Additionally, simulations are usually based on the theories and formulations. So, what is the meaning of differences between these two values? Is it reasonable?

Author Response

Point 1: Line 174 needs a reference instead of “authors' previous research“.

Response 1: According to the reviewers' comments, the expressions have been revised in the paper and reference has been cited, as detailed in line 174. 

Point 2:   Reference(s) should be included to Table caption(s) in Table 2 and 3.

Response 2: According to the reviewers' comments, the reference have been cited in Table 2 and Table 3.

Point 3: Some figures are not in good quality, for instant Figure 3.

Response 3: I am sorry, as Figure 3 is a cited image from the reference. There is no way to obtain the source files, so the quality of the image is not good.

Point 4: References should be rewritten correctly and in the same format. For example, No.1 (Van Zeebroeck, M.; ...); No.2 (Kafashan, J.; ...); No. 3 (incomplete the author name and format matter); and No. 7, 8, 9 (Yan, D;). No. 15 again. And using “, et al.” for No. 12, 14, 18. Yet, other relevant references could be cited and added.

Response 4: According to the reviewers' comments, the authors have revised the corresponding references and cited new ones, as detailed in the references section of the paper.

Point 5: Additionally, simulations are usually based on the theories and formulations. So, what is the meaning of differences between these two values? Is it reasonable?

Response 5: Simulation is similar to test in that the results of each simulation are not the same. Therefore, the mean value of the simulation results still deviates from the theoretical value.

This manuscript is a resubmission of an earlier submission. The following is a list of the peer review reports and author responses from that submission.

Round 1

Reviewer 1 Report

This paper analyzes the combined use of the DEM method and the MBD method for the validation of the soybean planting simulation. And different variables are applied to determine the positioning of the seed in the planting furrow.

- Being such a specific study, carried out with soybeans, perhaps this should be included in the title.

- I imagine that the results may be extended to other seeds of other crops, but nothing is said about it.

- Including SOIL in the keywords does not seem very justified. It is a concept too generic. Perhaps, it would be better to use a combined element as "planting-soil"

- Figure 1 speaks of a three-dimensional view; however it is a rather flat image, in 2D. Show an image in perspective. And leave well indicated the direction X of advance in sowing.

- Figure 2 shows two views of a mechanical assembly. In a standard representation, the projection system used and its symbol should be reflected. In European System below the plan view and above the elevation view.

   At least indicate in each diagram if it is front view or top view.

- Together with figure 6, perhaps a diagram of "the cover tension angle" should be included and indicate with respect to which direction it is measured.

- Lin.118: also indicate the length (or working width) of the tested cylinder.

- In the images of figure 8 the "opener" is not well observed.

- Lin. 128: It is said that the seeds were sown by hand... I don't understand

- Lin. 146 - RecurDyn and EDEM software is said to be used. It is not justified why this software is used and not another. Why is a specific CAD-CAM-CAE software not used for the simulation of dynamic mechanical assemblies?

- In fig. 11: in the figure caption a) b) and c) should be indicated with their corresponding definition.

- In fig, 14 appears the expression "after repressing".   Shouldn't it be "after compacting?

- On the line. 285 it is said that "...which is within the acceptable error range". I do not understand on what basis this is assured.  How much is an acceptable value?

- Perhaps a specific section on METHODOLOGY was missing, although I understand that the authors have covered it with subsections 2-3-4.

- In the CONCLUSION section, indicate for what other uses the described methodology of combination of DEM and MBD is applicable. And also the authors should talk about possible future developments to achieve greater precision.

- Perhaps too local references are seen...  Many authors of Chinese origin.  It would be necessary to try to reference something more from other nationalities.

Author Response

Response to Reviewer 1 Comments

This paper analyzes the combined use of the DEM method and the MBD method for the validation of the soybean planting simulation. And different variables are applied to determine the positioning of the seed in the planting furrow.

Point 1: Being such a specific study, carried out with soybeans, perhaps this should be included in the title.

Response 1: Based on the reviewer's comment, the title of the article was revised to "Test and simulation analysis of the working process of soybean seeding monomer".

Point 2: I imagine that the results may be extended to other seeds of other crops, but nothing is said about it.

Response 2: In this paper, only the working process of a seeding monomer for soybean seeds has been tested and simulated. Tests on other seeds and seeding machines have not been carried out, so the applicability of the test results to other crops requires further analysis and research.

Point 3: Including SOIL in the keywords does not seem very justified. It is a concept too generic. Perhaps, it would be better to use a combined element as "planting-soil"

Response 3: According to the reviewer's comments, the keyword "soil" was replaced with "seed-soil".

Point 4: Figure 1 speaks of a three-dimensional view; however it is a rather flat image, in 2D. Show an image in perspective. And leave well indicated the direction X of advance in sowing.

Response 4: Figure 1 has been revised in the article based on the comments of the reviewer.

Point 5: Figure 2 shows two views of a mechanical assembly. In a standard representation, the projection system used and its symbol should be reflected. In European System below the plan view and above the elevation view. At least indicate in each diagram if it is front view or top view. Together with figure 6, perhaps a diagram of "the cover tension angle" should be included and indicate with respect to which direction it is measured.

Response 5: According to the reviewer's comments, Figures 2 and 6 have been modified.

Point 6: Lin.118: also indicate the length (or working width) of the tested cylinder.

Response 6: According to the reviewer's comments, the working width of the tested cylinder was indicated.

Point 7: In the images of figure 8 the "opener" is not well observed.

Response 7: The display of the Opener is not very good due to the angle of the photo. The test device had been removed and could not be rephotographed.

Point 8: Lin. 128: It is said that the seeds were sown by hand... I don't understand

Response 8: It means to plant seeds artificially.

Point 9: Lin. 146 - RecurDyn and EDEM software is said to be used. It is not justified why this software is used and not another. Why is a specific CAD-CAM-CAE software not used for the simulation of dynamic mechanical assemblies?

Response 9: There are many software that can realize the coupling of DEM and MBD, this paper only selected the RecurDyn software. Therefore, there is no comparative analysis with other software.

Point 10: In fig. 11: in the figure caption a) b) and c) should be indicated with their corresponding definition.

Response 10: According to the reviewer's comments, Figure 11 has been modified.

Point 11: In fig, 14 appears the expression "after repressing".   Shouldn't it be "after compacting?

Response 11: According to the reviewer's comments, Figure 14 has been modified.

Point 12: On the line. 285 it is said that "...which is within the acceptable error range". I do not understand on what basis this is assured.  How much is an acceptable value?

Response 12: The expression is not rigorous. It was revised according to the reviewer's comments.

Point 13: Perhaps a specific section on METHODOLOGY was missing, although I understand that the authors have covered it with subsections 2-3-4.

Response 13: As the coupling method has already been described in subsections 2-3-4, no further summary is provided.

Point 14: In the CONCLUSION section, indicate for what other uses the described methodology of combination of DEM and MBD is applicable. And also the authors should talk about possible future developments to achieve greater precision.

Response 14: According to the reviewer's comments, the prospects for the application of the coupled DEM and MBD method are described in the conclusion section of the paper. Also, the conclusion section has been re-capitulated.

Point 15: Perhaps too local references are seen...  Many authors of Chinese origin.  It would be necessary to try to reference something more from other nationalities.

Response 15: In the field of agricultural engineering, there are few foreign studies on the coupling method of DEM and MBD, so there are relatively more references in China. In my future work, I will consult more relevant foreign references.

Reviewer 2 Report

The numerical simulation of the seeding monomer working process is presented in this manuscript. The simulation results appear to be in fair agreement with the experimental result or analytical value. However, there is little information about the referred experimental result or analytical value. Moreover, essential technical information is significantly lacking. The conclusion does not provide obvious archival value, as no new finding or suggestion of best practice is state. I suggest rejection to this submission and encourage the authors to resubmit upon a thoughtful and thorough revision.

Below are some typical issues.

Are Figures 2, 3 and 5 produced by the author(s) or adapted from published literature? If these figures are from published literature, the author(s) should provide reference in the figure caption.

Is Figure 6 produced by the author(s)? If yes, please provide date and location of the photo in the figure caption; if not, please provide reference in the figure caption.

Line 109: it seems there is a missing reference “[]”. Otherwise, remove “[]”.

I assume the photo in Figure 7 was produced by the author(s). If so, please provide date and location in the figure caption. Otherwise, please provide reference in the figure caption.

I suggest keeping either Figure 9 or Figure 1.

Line 157-159: description of software operation is not necessary in the main body of a manuscript. If the author(s) would like to provide software user instruction, they can provide a separate Supplemental Material document or appendix.

Line 167-168: it is not appropriate to simply say “according to the authors’ previous research” without a reference. Please provide reference.

Line 187-190: to be more effective in describing the directions, please consider adding the symbols in Figure 10 or Figure 11.

The resolution of Figure 11 is rather poor. Can the authors provide high-resolution images? Also, the figure caption should provide description of subfigure (a), (b), (c), and (d).

Line 195-197: I do not see how Figure 11(d) shows that the soybean seed particles are significantly displaced in the X-axis direction.

Figures 12 and 13: It has not been mentioned where the “experiment result” is from in those figures. If it is from prior published work, please provide reference. If it is part of the present work, please clarify.

Figure 20: There is no reference or description of the “theoretical value”. Please provide an equation and reference.

Line 296-297: (1) is not considered as a piece of conclusion, as no conclusive information is provided.

Line 298-300: (2) is also not providing any conclusive information.

Line 301-308: (3) can be split into two items in the list.

Author Response

Response to Reviewer 2 Comments

Point 1: Are Figures 2, 3 and 5 produced by the author(s) or adapted from published literature? If these figures are from published literature, the author(s) should provide reference in the figure caption.

Response 1: Based on the comments of the reviewer, relevant reference was cited to Figures 2, 3 and 5.

Point 2: Is Figure 6 produced by the author(s)? If yes, please provide date and location of the photo in the figure caption; if not, please provide reference in the figure caption.

Response 2: The Figure 6 is produced by the author. The paper has been revised in accord with the comments of the reviewer. 

Point 3: Line 109: it seems there is a missing reference “[]”. Otherwise, remove “[]”.

Response 3: The reference was missing. The paper has been revised in accord with the comments of the reviewer. 

Point 4: I assume the photo in Figure 7 was produced by the author(s). If so, please provide date and location in the figure caption. Otherwise, please provide reference in the figure caption.

Response 4: The Figure 7 is produced by the authors. The paper has been revised in accord with the comments of the reviewer. 

Point 5: I suggest keeping either Figure 9 or Figure 1.

Response 5: Based on the reviewer's comments, the authors have removed the Figure 9.

Point 6: Line 157-159: description of software operation is not necessary in the main body of a manuscript. If the author(s) would like to provide software user instruction, they can provide a separate Supplemental Material document or appendix.

Response 6 : Based on the reviewer's comments, the authors have removed the relevant content.

Point 7: Line 167-168: it is not appropriate to simply say “according to the authors’ previous research” without a reference. Please provide reference.

Response 7: The paper was revised according to the reviewer's comments.

Point 8: Line 187-190: to be more effective in describing the directions, please consider adding the symbols in Figure 10 or Figure 11.

Response 8: According to the reviewer's comments, the symbols were added in Figure 10 or Figure 11.

Point 9: The resolution of Figure 11 is rather poor. Can the authors provide high-resolution images? Also, the figure caption should provide description of subfigure (a), (b), (c), and (d).

Response 9: Figure 11 is a screenshot taken in the software, so the resolution is low. The figure caption has been revised according to the comments of the reviewer.

Point 10: Line 195-197: I do not see how Figure 11(d) shows that the soybean seed particles are significantly displaced in the X-axis direction.

Response 10: The position of the Geometry Bin is fixed. The distance of the soybean seed particle from the edge of the Geometry Bin is distinctly different.

Point 11: Figures 12 and 13: It has not been mentioned where the “experiment result” is from in those figures. If it is from prior published work, please provide reference. If it is part of the present work, please clarify.

Response 11: Based on the comments of the reviewer, the relevant reference to the experiment result of the previous work is cited.

Point 12: Figure 20: There is no reference or description of the “theoretical value”. Please provide an equation and reference.

Response 12: The theoretical values are obtained by simple calculations and there is no relevant reference.

Point 13: Line 296-297: (1) is not considered as a piece of conclusion, as no conclusive information is provided.

Response 13: Based on the comments of the reviewer, the conclusion (1) was deated.

Point 14: Line 298-300: (2) is also not providing any conclusive information.

Response 14: Based on the comments of the reviewer, the conclusion (2) was deated.

Point 15: Line 301-308: (3) can be split into two items in the list.

Response 15: Based on the comments of the reviewer, the conclusion (3) was split into two items. The conclusions are also re-summarised.

Reviewer 3 Report

Very good work by authors on this paper. 

I have few general comments on usage of DEM and few points to add in discussion/future sections if not feasible to add in manuscript.

1. Using clumping of particles to have a bit irregular shape of particles to have a mixture of particles.

2. Running simulation at 3 different particle sizes, 10, 15 and 20 mm to expand your work's application in other DEM application. (This is not a mandatory thing) but would help research community and get you cited more. 

3. One of the observation for DEM simulations is that they are sensitive to loading rate, meaning they will perform lower than experimental at lower speed, and higher at higher speeds for experiment. I can see such behavior not exactly similar to what I explained. So will be great to have more runs to have better understanding. 

4. Explanation on why a specific point is falling behind or more than experimental setup. 

5. Did you measure properties of experimental soil and used it for DEM simulation or you used literature values for DEM? 

Author Response

Response to Reviewer 3 Comments

I have few general comments on usage of DEM and few points to add in discussion/future sections if not feasible to add in manuscript.

Point 1: Using clumping of particles to have a bit irregular shape of particles to have a mixture of particles.

Response 1: The concluding section of the paper has been revised based on the comments of the reviewer.

Point 2: Running simulation at 3 different particle sizes, 10, 15 and 20 mm to expand your work's application in other DEM application. (This is not a mandatory thing) but would help research community and get you cited more. 

Response 2: The reviewer's comments are very accurate. In future research work, different particle sizes can be used when modelling soil particles

Point 3: One of the observation for DEM simulations is that they are sensitive to loading rate, meaning they will perform lower than experimental at lower speed, and higher at higher speeds for experiment. I can see such behavior not exactly similar to what I explained. So will be great to have more runs to have better understanding. 

Response 3: As the parameters used in the simulation are either measured experimentally or from the relevant literature, the EDEM simulation is in fact consistent with the test. However it is true that more simulation runs can be done to increase the accuracy of the results

Point 4: Explanation on why a specific point is falling behind or more than experimental setup.

Response 4: In fact, both simulation tests and trials are somewhat stochastic in nature. Therefore both the test and simulation results fluctuate, which means that sometimes the simulation results are higher than the test results, and sometimes the simulation results are lower than the test results.

Point 5: Did you measure properties of experimental soil and used it for DEM simulation or you used literature values for DEM? 

Response 5: The parameters of the soil particles in the test were measured experimentally.
